# Effects of Heat Treatment on Microstructure and Mechanical Properties of Weldable Al–Mg–Zn–Sc Alloy with High Strength and Ductility

**DOI:** 10.3390/ma16155435

**Published:** 2023-08-03

**Authors:** Long Jiang, Zhifeng Zhang, Yuelong Bai, Weimin Mao

**Affiliations:** 1National Engineering Research Center for Non-Ferrous Metal Composites, China GRINM Group Co., Ltd., Beijing 100088, China; b20190171@xs.ustb.edu.cn (L.J.); baiyuelong@grinm.com (Y.B.); 2School of Materials Science and Engineering, University of Science and Technology Beijing, Beijing 100083, China; mao_wm@ustb.edu.cn; 3Grinm Metal Composites Technology Co., Ltd., Beijing 100088, China; 4General Research Institute for Nonferrous Metals, Beijing 100088, China

**Keywords:** Al–Mg–Zn–Sc alloys, microstructure evolution, mechanical properties, heat treatments

## Abstract

A weldable Al–Mg–Zn–Sc alloy was produced using vacuum induction melting and an argon-protected casting method to achieve high strength and ductility, and the effects of heat treatment on the microstructure evolution and mechanical properties of Al–Mg–Zn–Sc alloys were comparatively investigated. The results reveal that fine equiaxed grains with an average grain size of 40 μm in an as-cast Al–Mg–Zn–Sc alloy change little after heat treatments, bringing about a grain-boundary strengthening of 46.1 MPa. The coarse T-Mg_32_(Al, Zn)_49_ phases at grain boundaries are completely dissolved into the matrix through solid-solution treatment, and T-Mg_32_(Al, Zn)_49_ with diameters ranging from 10 to 25 nm and Al_3_Sc with diameters ranging from 5 to 20 nm gradually precipitate during the artificial aging process. The Mg solid solubility is 4.67% in the as-cast Al–Mg–Zn–Sc alloy, and it increased to 5.33% after solid-solution treatment and dramatically decreased to 4.15% after post-aging treatment. The contributions of solid-solution strengthening to as-cast, post-solid-solution and post-aging Al–Mg–Zn–Sc alloys are 78.2 MPa, 85.4 MPa and 72.3 MPa, respectively. The precipitation strengthening of the post-aging alloy is 49.7 MPa, which is an increase of 21% in comparison to that of both as-cast and post-solid-solution alloys. The alloy achieves an optimal tensile strength of 355.3 MPa, yield strength of 175 MPa and elongation of 22% after undergoing solid-solution treatment.

## 1. Introduction

5xxx Al-Mg alloys provide a high degree of weldability and corrosion resistance with a medium level of strength due to their high Mg content [1,2,3]. However, Al–Mg alloys with high Mg content tend to form β-Al_3_Mg_2_ phases, resulting in the deterioration of mechanical properties [4]. Moreover, nonbeneficial kinetics impede precipitation hardening by β-Al_3_Mg_2_, which renders Al–Mg alloys non-heat-treatable [5]. For the traditional Al–Mg alloys, the inability to strengthen through heat treatment poses a significant challenge for their widespread application. Thus, it is necessary to develop new strategies to solve the above problems and improve the mechanical properties of Al–Mg alloys.

Some attempts have been made to eliminate dendrites, refine grains and introduce secondary phases in order to enhance the mechanical properties of Al–Mg alloys [6]. In addition to the conventional strategy of optimizing casting process parameters [7,8,9], adding alloy elements for boosting the precipitation hardening of Al–Mg alloys is a promising attempt to enhance their mechanical properties. Ratchev et al. [10] found that adding Cu was sufficient to induce rapid hardening due to the formation of the precursor to the S-Al_2_CuMg phase but at the expense of the poor corrosion resistance of such alloys [11]. Zn-modified Al–Mg alloys lead to the formation of the T-Mg_32_(Al, Zn)_49_ phases, which exhibit improved corrosion resistance and increased strength caused by precipitation hardening [12,13], in spite of the trade-off between precipitation and solid-solution strengthening. Newer grades of 5xxx alloys have additions of Sc to improve mechanical properties. Taendl et al. [14] demonstrated that an effective aging treatment can induce the diffusion of Sc into the alloy to form Al_3_Sc nanoprecipitates and thus harden the alloy during a controlled heat treatment. While Al_3_Sc precipitates are more likely to occur along grain boundaries, the inhomogeneous distribution of the secondary phases has a significant impact on strength and ductility [15].

The precipitation of Mg-rich nanoprecipitates in Al–Mg alloys has been manipulated for the strengthening of Cu/Zn-modified Al–Mg alloys by means of appropriate heat treatments [16,17]. However, these high-Mg nanoprecipitates are obtained at the cost of a large proportion of solute Mg atoms in the α–Al matrix, and such age-strengthened Al–Mg alloys exhibit a reduced ductility [18]. In order to achieve an enhanced combination of strength and ductility, a high-Mg Al–Mg–Zn–Sc alloy was designed in our study, which simultaneously includes high yield strength and elongation. However, so far little work has been done about the detailed information of strength contribution and microstructure evolution of this Al–Mg–Zn–Sc alloy in different heat treatments. It is also important to clarify the hedging relationship between solid-solution strengthening and precipitation strengthening.

The aim of this study is to investigate the microstructure evolution and mechanical properties of the Al–Mg–Zn–Sc alloys under solid-solution treatment and solution-aging treatment. The Mg solid solubility at different heat treatments is quantitatively calculated by X-ray diffractometer (XRD); dual-nanoprecipitation phases of Al_3_Sc and T-Mg_32_(Al, Zn)_49_ after post-aging are analyzed by transmission electron microscope (TEM); and a quantitative elaboration on the strengthening mechanism is discussed.

## 2. Experimental Section

The experimental Al–Mg–Zn–Sc alloy used in this work was prepared by melting in a vacuum and casting in a steel mold under the protection of argon. The chemical compositions of the studied alloy are listed in Table 1.

In this study, two sets of heat treatment experiments were conducted on the as-cast Al–Mg–Zn–Sc alloy, and the schematic diagrams of the heat treatment procedures are shown in Figure 1. Both samples were solid-solution treated with an air furnace at 495 °C for 50 min and then quenched to room temperature in water. A uniform supersaturated solid solution was obtained in this step. One of the quenched samples was artificially aged at 120 °C for 48 h. For clarity, the as-cast sample is called “AC”, the solid-solution-treatment sample is called “ST” and the artificial-aging sample is called “AA”.

Microstructural characterization was performed using optical microscopy (OM Axiovert 200 MAT ZEISS), scanning electron microscopy (SEM JSM. 7001F) equipped with energy dispersive spectroscopy (EDS) and a transmission electron microscope (TEM Tecnal G^2^ F20 S-TWIN). X-ray diffraction (XRD D8 ADVANCE) was performed to identify the phase types at a scan step of 0.5°/min. Tensile specimens of 35 mm gauge length and 5 mm diameter were machined from the ingot along the casting direction. Tensile tests were performed on the CSS-4100 electronic universal material testing machine with a strain rate of 1.5 × 10^−3^ s^−1^.

## 3. Results

### 3.1. Microstructure Evolution

The microstructure of the Al–Mg–Zn–Sc alloy after different heat treatments is shown in Figure 2. Fine and equiaxed grains are found in all Al–Mg–Zn–Sc alloy samples under various conditions. The average grain size is approximately 40 μm. The microstructural evolution during heat treatment is shown in Figure 2d–f. As shown in the OM image of Figure 2d, the as-cast microstructure of the Al–Mg–Zn–Sc alloy is composed of α-Al phases, a large fraction of grain boundary (GB) phases and a small fraction of second phases. After solid-solution treatment, the coarse GB phases and discontinuous GB phases are almost totally solubilized into the matrix, promoting the dispersion of the clusterlike second phases into the granular second phases in high-temperature solid-solution treatment, as shown in Figure 2e [19]. As for ST + AA samples, the nanophases progressively precipitated during the artificial aging process (TEM images of Figure 5). However, this process caused a reprecipitation and coarsening of GB phases, as well as an increasing tendency of the granular second phases to form clusterlike second phases, as shown in Figure 2f.

Figure 3 depicts the SEM and TEM images of the as-cast Al–Mg–Zn–Sc alloy. Four phases with different contrasts are observed in a backscattered electrons (BSE) image of the AC alloy, as shown in Figure 3a. According to the EDS mapping results, Mg and Zn are deposited at grain boundaries, and Al is slightly depleted, which implies that the GB phases are Al–Mg–Zn phases. In the bright-white phase marked with “spectrum 2” shown in Figure 3a, which is an Fe-rich and Mn-rich phase, the Fe element can be derived from the raw materials to form Al_6_(Fe, Mn) phases [20]. The gray phases marked with “spectrum 3” and “spectrum 4” are viewed as Al–Sc–Ti phases due to the overlap of the distribution of Sc and Ti elements, as displayed in Figure 3a. The Al–Mg–Zn phases are confirmed to be T-Mg_32_(Al, Zn)_49_ phases in combination with the TEM analysis in Figure 3b. The T phase shows a semicontinuous irregular block morphology with narrow dimension size less than 2 μm in the bright-field (BF) TEM image. Figure 3c presents a high-angle annular dark-field (HAADF) image of a micron-size tetragonal primary phase. As can be seen from the selected area electron diffraction (SAED) pattern, the primary phase exhibits an L12 ordered superlattice of Al_3_Sc diffraction spots with the central grain oriented parallel to the axis of the [112] zone. Ti can replace up to the Sc to form Al_3_(Sc, Ti) phases while retaining the L12 structure [21]. There are fine and spherical precipitates with a size of 5–20 nm distributed in the α–Al matrix in the dark-field (DF) TEM image, as shown in Figure 3d. While performing fast Fourier transform (FFT) on the high-resolution transmission electron microscopy (HRTEM) image, it can be seen that the diffraction spots of Al_3_Sc are located in the 1/2 (200)_Al_ and (022)_Al_ positions. Meanwhile, the Al_3_Sc phase and the α–Al matrix have the orientation relations of [011]_Al3Sc_//[011]_Al_.

SEM and TEM characterizations of the solid-solution-treated Al–Mg–Zn–Sc alloy are shown in Figure 4. As can be seen in Figure 4a, there are no obvious Al–Mg–Zn phases to be observed in the BSE image. This demonstrates that Mg and Zn atoms are dissolved into the α–Al matrix after solid-solution treatment. Based on the results of EDS analysis, the Al_6_(Fe, Mn) phases with less content did not disappear in the ST alloy, since the solid-solution temperature is much lower than the dissolution temperature of Al_6_(Fe, Mn) [22]. Figure 4b revealed these tetragonal primary Al_3_(Sc, Ti) phases distributed along the Al grain boundaries, which are similar to those in the AC sample (Figure 3a). The Al_3_Sc decomposition is limited at high solid-solution temperature, but this can inhibit the coarsening of nanoscale Al_3_Sc precipitates, as shown in Figure 4c [19].

Figure 5a shows that the BSE image of the micron-scale second phase of the Al–Mg–Zn–Sc alloy after artificial aging is similar to that of the ST sample, as characterized by SEM. However, the volume fraction and dispersion of precipitates increased significantly after aging, as shown in Figure 5b. Meanwhile, the nanoscale phase shows two morphologies, beanlike (shown by green arrow) and squarelike (shown by red arrow), respectively. By analyzing the diffraction spots of the FFT in Figure 5e, the tiny beanlike precipitates were identified as Al_3_Sc phases. As shown in Figure 5g–i, new squarelike precipitated phases after the aging process were indexed as T-Mg_32_(Al, Zn)_49_ phases using high-resolution transmission electron microscopy (HRTEM), fast Fourier transform (FFT) and inverse fast Fourier transform (IFFT). The above results show that the dual-nanoprecipitation of the Al–Mg–Zn–Sc alloy can be induced after artificial aging treatment. The nanoprecipitations included T-Mg_32_(Al, Zn)_49_ 10–25 nm in diameter and Al_3_Sc 5–20 nm in diameter, as shown in Figure 5b,c.

### 3.2. Characterization of the Solid Solubility

XRD patterns of the Al–Mg–Zn–Sc alloy after various heat treatments are seen in Figure 6. The reflections (111), (200), (220), (311) and (222) of the aluminum appeared in the XRD pattern. The obvious offset to smaller diffraction angles of the XRD diffraction peaks is noticed for all Al–Mg–Zn–Sc alloy samples with respect to the pure aluminum. The offset of diffraction peaks is due to the variation in the lattice constant of the aluminum caused by the solid solution of Mg in the matrix α–Al [23]. The literature shows that for each 1 at% increase in the solid solution of Mg in Al, the lattice constant of Al increases by 4.6 × 10^−4^ nm [24]. The results of the lattice constant and solid solubility calculation of the studied Al–Mg–(Zn–Sc) alloy by XRD analysis are given in Table 2. The solid solubility of AC, ST and ST + AA samples were 4.67%, 5.33% and 4.15%, respectively. The Al–Mg–Zn–Sc alloy after solid-solution alloy has the largest solid solubility, which increases by 12.4% compared to the as-cast alloy and by 22.1% compared to the post-aging alloy.

### 3.3. Mechanical Properties

Figure 7a illustrates the engineering stress–strain curves of the Al–Mg–Zn–Sc alloy studied after different heat treatments. The corresponding strengths and elongations are shown in Figure 7b. The strength and elongation of the alloy after solid-solution treatment are both significantly higher than those of the alloy as-cast and after aging. In comparison to the AC sample, the tensile strength of ST is 355.3 MPa, which is a 6.9% increase. ST provides an increase of 10.3 MPa and achieves a yield strength of 175 MPa, while the elongation increases from 15.5% to 22%. After the artificial aging treatment, the ST + AA sample exhibits a weakening response compared to the ST sample. In particular, the yield strength is reduced by 7.6% compared to that of the ST sample. Additionally, ST + AA has the lowest yield strength, with a reduction of 13.3 MPa and 3 MPa compared to that of ST and AC, respectively.

## 4. Discussion

In this study, the Al–Mg–Zn–Sc alloy was obtained by metal mold casting. It exhibits fine equiaxed grains (Figure 2a), multiple dispersed phases of Al_3_Sc (Figure 3d) and high Mg solid solubility (Table 2) in as-cast Al–Mg–Zn–Sc alloy. These beneficial effects together enhance the mechanical properties of the Al–Mg–Zn–Sc alloy (Figure 7). Meanwhile, primary Al_3_(Sc, Ti) phases act as heterogeneous nuclei of the matrix during the solidification process and strongly refine the grains (Figure 2c). Moreover, the formation of micron T-Mg_32_(Al, Zn)_49_ phases along the grain boundaries dramatically enhance the corrosion resistance [25]. The expectation of microstructure evolution during heat treatment is plotted in Figure 8. After solid-solution treatment, the coarsening of the T-Mg_32_(Al, Zn)_49_ phases at the grain boundaries dissolve into α–Al matrix, leading to the supersaturated solid solution (SSSS) of Mg, as seen in Figure 4a. According to the previous studies of Al–Mg–Zn alloy, it was confirmed that the supersaturated vacancies are attached to Mg atoms rather than Zn atoms after water quenching owing to the stronger Mg-to-vacancy binding energy [5]. In addition, Mg atoms are 12% larger than Al atoms, whereas Zn atoms are smaller by 2.8% in size; therefore, the solid-solution effect mainly comes from the solid solubility of Mg atoms in Al [26]. The dissolution of primary Al_3_(Sc, Ti) phases and nano-Al_3_Sc precipitates are limited by the fact that the solid-solution temperature is much lower than the dissolution temperature of Al_3_(Sc, Ti) and Al_3_Sc phases (Figure 3a) [27]. When artificial aging is performed on the quenched sample, there is almost no strengthening effect compared to the as-cast alloy (Figure 7b). Affirmatively, some Al_3_Sc phases were precipitated from the α–Al matrix of as-cast Al–Mg–Zn–Sc with oversaturated Sc under the aging condition of 120 °C/48 h. However, overaging also leads to the aggregation and growth of the Al_3_Sc phases, as shown in Figure 8. According to the study of precipitation behavior in Zn-modified Al–Mg alloy at aging, there exists a precipitation sequence described as SSSS → GP-zone → T′ → T-Mg_32_(Al, Zn)_49_ [27]. Consequently, the precipitation of nanoscaled T phase improved the age-hardening behavior of Al–Mg–Zn alloys [13]. As analyzed in Figure 5b,c, the average volume fraction of nano-T-Mg_32_(Al, Zn)_49_ is 0.1%, causing only a small contribution to the precipitation hardening effect (Figure 7b). According to the analysis above, the strength and elongation of the Al–Mg–Zn–Sc alloy can be improved by proper solid-solution treatment (Figure 7), while in the aging treatment of the alloy, there is a contradictory strengthening effect with increased precipitation strengthening and reduced solid-solution strengthening (Table 2).

To elucidate the effectiveness of the heat treatment, the strengthening contributions coming from grain boundaries, solid soluble atoms and precipitates are quantified in the Al–Mg–Zn–Sc alloy. As shown in Figure 8, the yield strength (*σ_y_*) of the studied Al–Mg–Zn–Sc alloy can be calculated from grain boundary strengthening (*σ_GB_*), solid-solution strengthening (*σ_ss_*) and precipitation strengthening (*σ_p_*).

Since all the Al–Mg–Zn–Sc alloys show an almost identical average grain size of 40 μm (Figure 2), the change in grain boundary strengthening should be insignificant. Grain boundary strengthening arising from Al_3_(Sc, Ti) heterogeneous nucleation points is thought to be equivalent before and after heat treatment. The relationship between *σ_GB_* and the average grain size (*d*) of the alloys is in accordance with the Hall–Petch equation [13], as described below:(1)σGB=σ0+kd−12
where $\sigma_{0}$ (19.2 MPa) is the lattice intrinsic resistance to dislocation motion [28]. *k* is a coefficient representing relative grain boundary strengthening contribution (0.17 MPa⋅m^0.5^) [29]. *d* is the average size of the grains taken as 40 μm. The increase in yield strength due to grain boundary strengthening is thus calculated to be approximately 46.1 MPa.

Of all the strengthening factors, solid-solution strengthening is considered to be the most important strengthening mechanism governing the overall strength of Al–Mg–Zn–Sc alloys. The solid solubility of Mg in the Al–Mg–Zn–Sc alloy changed significantly both before and after heat treatment (as detailed in Table 2). This increase in yield strength can be roughly calculated by the following equation relating the number of solute atoms to the increase in strength [30]:(2)$$\sigma_{ss}=HC_{Mg}^{n}$$
where *H* and *n* are material related constants with values of 25–28 MPa (wt% Mg)^−n^ and 2/3, respectively [31,32]. *C* denotes the number of solute atoms of Mg (wt%). As a result, the *σ_ss_* values for the as-cast, post-solid solution and post-aging Al–Mg–Zn–Sc alloys are 78.2 MPa, 85.4 MPa and 72.3 MPa.

In our study, precipitation enhancement included the contributions from different types of precipitation. The volume fraction of Al_3_Sc precipitates is 0.28% for both as-cast and post-solid solution Al–Mg–Zn–Sc alloy, as observed in Figure 3d and Figure 4c, suggesting that both have the same precipitation-intensifying effect. Additionally, given the significant amounts of Al_3_Sc (volume fraction of 0.15%) and T-Mg_32_(Al, Zn)_49_ (volume fraction of 0.1%) in the post-aging Al–Mg–Zn–Sc alloy, it is essential to account for the contribution of dual precipitation strengthening in the Al–Mg–Zn–Sc alloy. The contribution of the Al_3_Sc and T-Mg_32_(Al, Zn)_49_ phases to the total strength can be calculated using the Ashby–Orowan equation [33]:(3)$$\sigma_{Orowan}=M\frac{0.4Gb}{\pi \lambda }\frac{ln(\frac{2\bar{r}}{b})}{\sqrt{1-\nu }}$$
where $\bar{r\;}$ is the mean radius of the precipitate. *M* = 3.06 is the Taylor factor [34]. *ν* is the Poisson’s ratio of this alloy (0.32). *G* is the shear modulus (about 26.9 GPa [35]). *b* is the Burgers vector of Al (0.286 nm). *λ* is the precipitate edge spacing, which can be calculated by the following equation (the volume fraction (*f*)) [34]:(4)$$\lambda =\bar{r}{\left(\frac{2\pi }{3f}\right)}^{1/2}$$

The calculation results show that the contribution of the precipitation strengthening was around 41.1 MPa, 41.1 MPa and 49.7 MPa for the as-cast, post-solid-solution and post-aging Al–Mg–Zn–Sc alloys, respectively. The Al–Mg–Zn–Sc alloy demonstrates a less significant contribution of precipitation strengthening through the aging treatment.

As shown in Figure 9, the strengthening contribution to the yield strength was calculated to be approximately 165.4 MPa, 172.6 MPa and 168.1 MPa for the as-cast, post-solid-solution and post-aging Al–Mg–Zn–Sc alloys, respectively. After solid-solution treatment, the significant strengthening of the Al–Mg–Zn–Sc alloy originates from solid-solution strengthening. The strengthening contribution of solution Mg in the Al–Mg–Zn–Sc alloy improves by 9.2% compared to the as-cast alloy. However, in the post-aging alloy, the contribution of solid-solution strengthening decreased by 7.5% and the contribution of precipitation strengthening increased by 21% compared to the as-cast alloy. There was a difference between the theoretical calculations and experimental results for the post-aging Al–Mg–Zn–Sc alloy. This calculation result is not a reflection of the age-hardening effect. However, the calculation results show that there is a contradictory strengthening effect with increased precipitation strengthening and reduced solid-solution strengthening in the post-aging alloy, which is the same as the experimental results. In the case of Al–Mg–Zn–Sc alloys, it is an appropriate way to improve the mechanical properties by increasing the solid solution of Mg in the alloy.

## 5. Conclusions

In this study, the microstructure evolution, mechanical properties, solid-solution effect and aging behavior of Al–Mg–Zn–Sc alloys produced by vacuum melting and argon-protected casting were investigated. The strengthening mechanisms induced by the microstructure were elucidated. The main conclusions drawn are as follows.

(1)As-cast Al–Mg–Zn–Sc alloy has fine equiaxed grains 40 μm in size with a high Mg solid solubility of 4.67% and large volume fraction of 0.15% Al_3_Sc nanoprecipitations. The solid solubility of Mg is increased to 5.33% after solid-solution treatment and dramatically decreased to 4.15% after post-aging treatment, but a dual nanoprecipitation including T-Mg_32_(Al, Zn)_49_ with diameters of 10 to 25 nm and Al_3_Sc with diameters of 5 to 20 nm is observed.(2)The solid-solution strengthening contributions to the as-cast, post-solid-solution and post-aging Al–Mg–Zn–Sc alloys are 78.2 MPa, 85.4 MPa and 72.3 MPa, respectively. The precipitation strengthening of the post-aging alloy is 49.7 MPa, which is an increase of 21% in comparison to that of both as-cast and post-solid-solution alloy. The tensile strength of 355.3 MPa, yield strength of 175 MPa and elongation of 22% for the alloy are optimally obtained after solid-solution treatment.(3)Solid-solution strengthening is the main strengthening mechanism controlling the ultimate strength of the Al–Mg–Zn–Sc alloys. Contradictory strengthening effects exists in the post-aging alloy with increased precipitation strengthening and reduced solid-solution strengthening. It is an effective way to improve the strengthening effect by increasing the solid solution of Mg in the Al–Mg–Zn–Sc alloys.

## Figures and Tables

**Figure 1 materials-16-05435-f001:**
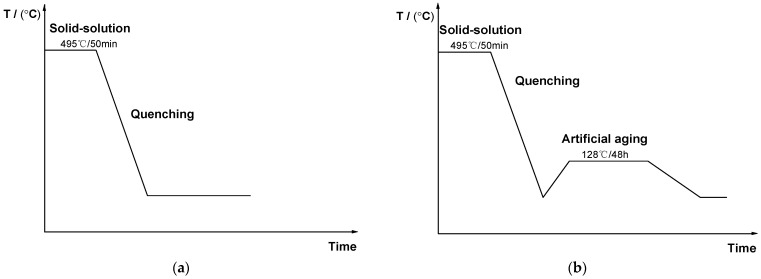
Schematic diagrams of two heat treatments for Al–Mg–Zn–Sc alloy: (**a**) ST sample; (**b**) ST + AA sample.

**Figure 2 materials-16-05435-f002:**
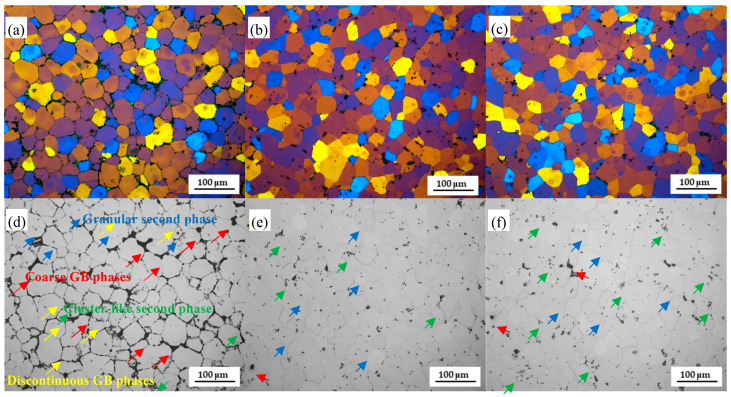
Microstructure of Al–Mg–Zn–Sc alloy: (**a**,**d**) AC sample; (**b**,**e**) ST sample; and (**c**,**f**) ST + AA sample.

**Figure 3 materials-16-05435-f003:**
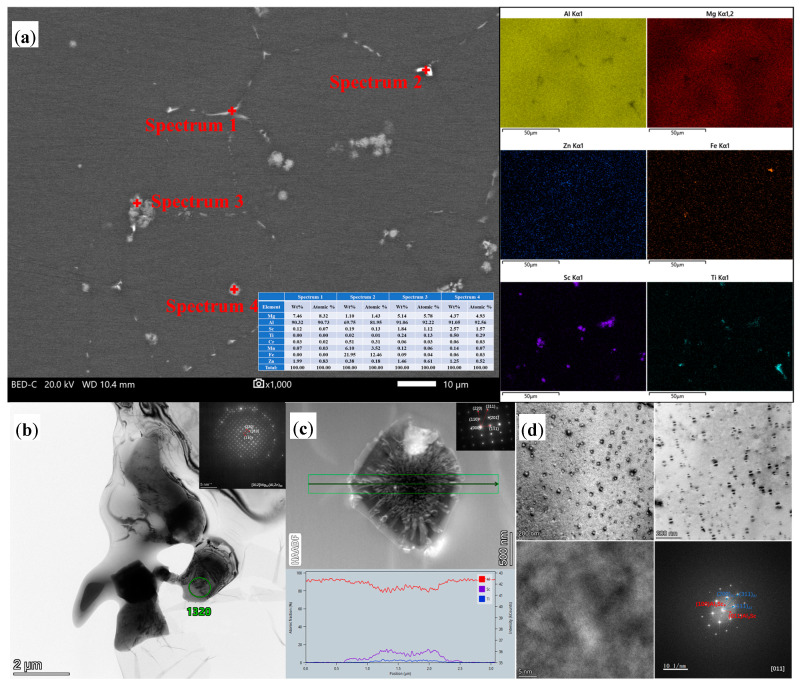
SEM and TEM images of as-cast (AC) Al–Mg–Zn–Sc alloy: (**a**) BSE image and corresponding EDS mapping; (**b**) BF-TEM image and SAD pattern of T-Mg_32_(Al, Zn)_49_ phase; (**c**) HAADF image and SAED pattern of Al_3_(Sc, Ti) phase; and (**d**) DF-TEM, BF-TEM and HRTEM images and corresponding FFT of nano-Al_3_Sc particles.

**Figure 4 materials-16-05435-f004:**
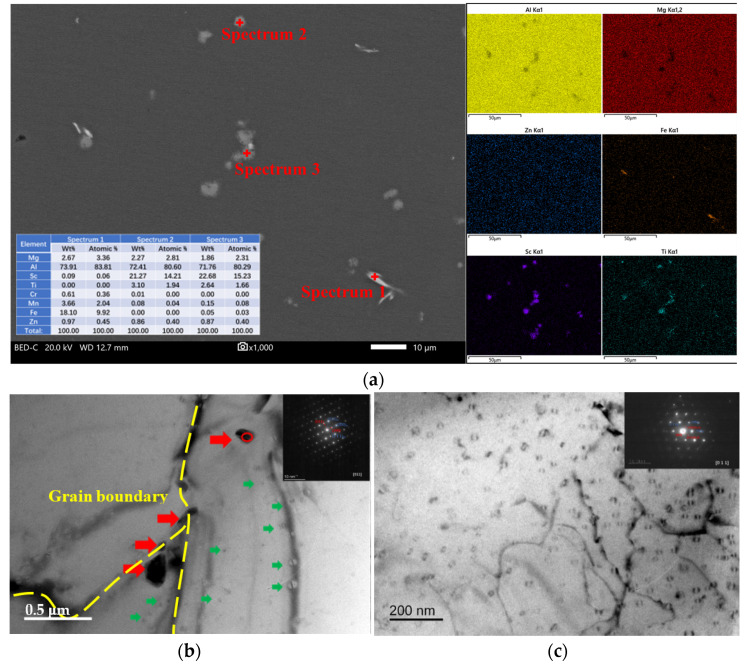
SEM and TEM images of solid-solution-treatment (ST) Al–Mg–Zn–Sc alloy: (**a**) BSE image and corresponding EDS mapping; (**b**) BF-TEM image (red arrows for Al_3_(Sc, Ti) phases and green arrows for nano Al_3_Sc phases) and SAED pattern of Al_3_(Sc, Ti) phase; and (**c**) BF-TEM image and SAED pattern of nano-Al_3_Sc particles.

**Figure 5 materials-16-05435-f005:**
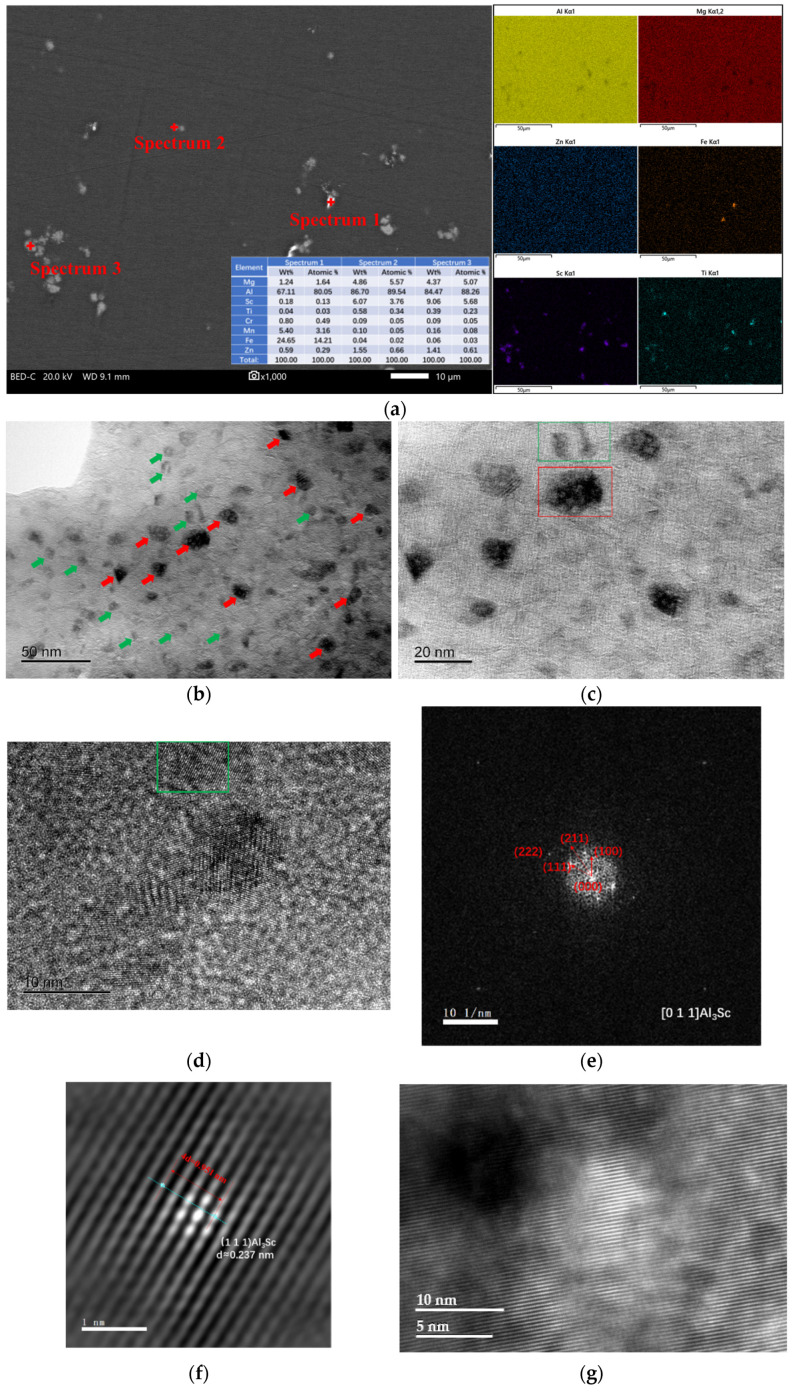
SEM and TEM images of artificially aged (AA) Al–Mg–Zn–Sc alloy: (**a**) BSE image of two phases (beanlike and squarelike phases are shown by green arrows and red arrows, respectively), and corresponding EDS mapping; (**b**) BF-TEM image of two types of nanoparticles (green box for beanlike phase and red box for squarelike phase); (**c**) high magnification nanoparticles in (**b**); (**d**) HRTEM images of nano-Al_3_Sc particles; (**e**) corresponding FFT of (**d**); (**f**) corresponding IFFT image of (**d**) after applying Bragg filter to Al_3_Sc FFT spots; (**g**) HRTEM images of nano-T-Mg_32_(Al, Zn)_49_ particles; (**h**) corresponding FFT of (**g**); and (**i**) corresponding IFFT image of (**g**) after applying Bragg filter to FFT spots.

**Figure 6 materials-16-05435-f006:**
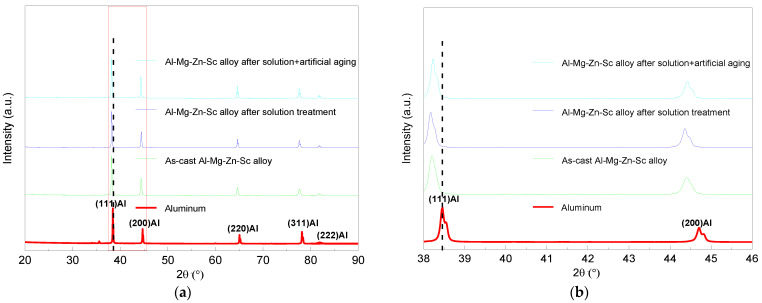
XRD patterns of the studied Al–Mg–Zn–Sc alloy after various heat treatments: (**a**) 2 θ (20°–90°) and (**b**) 2 θ (38°–46°).

**Figure 7 materials-16-05435-f007:**
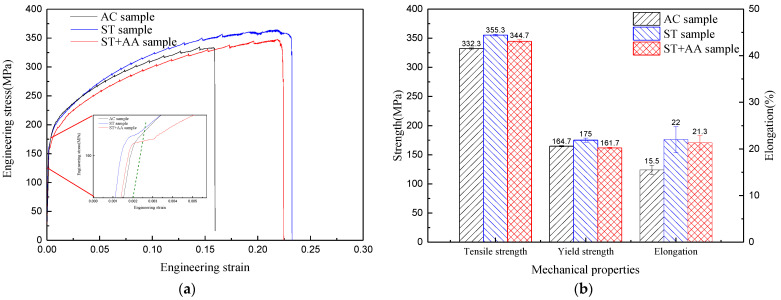
Mechanical properties of the studied Al–Mg–Zn–Sc alloys: (**a**) the engineering stress–strain curves and (**b**) mechanical properties.

**Figure 8 materials-16-05435-f008:**
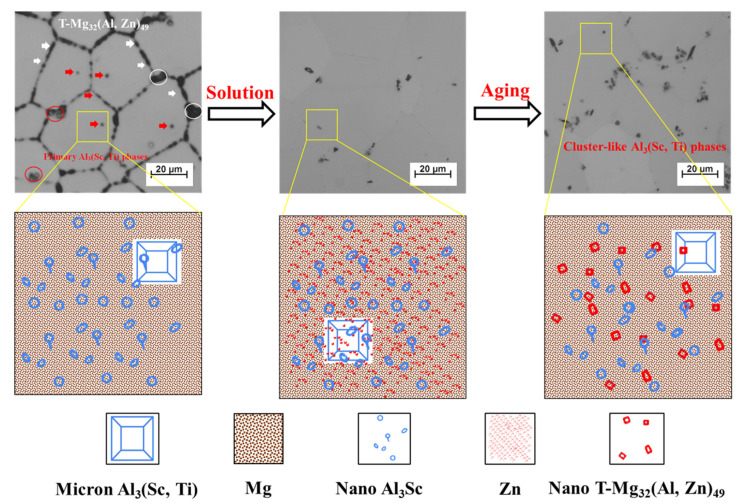
Schematic diagram of microstructure evolution of studied Al–Mg–Zn–Sc alloy after various heat treatments.

**Figure 9 materials-16-05435-f009:**
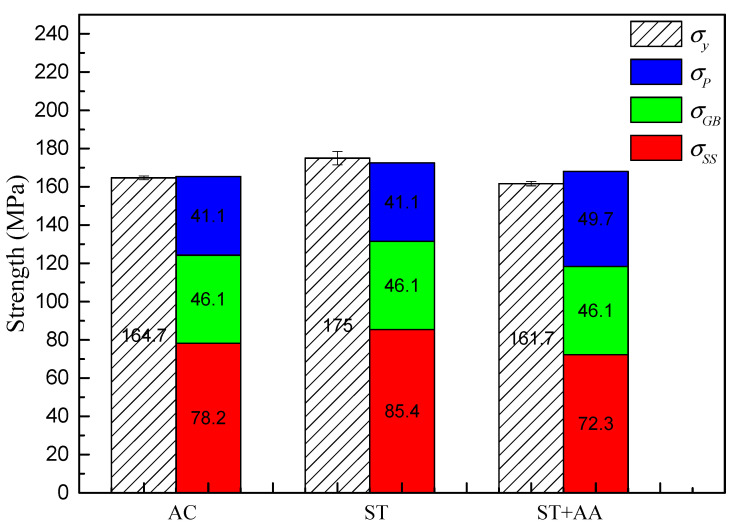
Comparison of calculated and experimental yield strength of Al–Mg–Zn–Sc alloy after various heat treatments.

**Table 1 materials-16-05435-t001:** Chemical compositions of the studied Al–Mg–Zn–Sc alloy (wt.%).

Elements	Mg	Zn	Sc	Mn	Cr	Ti	Fe	Si	Al
Al–Mg–Zn–Sc	5.45	1.49	0.40	0.152	0.062	0.103	0.022	0.008	Bal.

**Table 2 materials-16-05435-t002:** The results of the lattice constant and solid solubility calculation of the studied Al–Mg–(Zn–Sc) alloy by XRD analysis.

Sample	Lattice Parameter/nm	Crystal Indices		Solid Solubility/%
		(111)	(200)	
Pure aluminum	*d*	0.23379	0.20247	/
	*a*	0.40494	0.40494	
	Average value of *a*	0.40494		
AC	*d*	0.23513	0.20343	4.67
	*a*	0.40726	0.40692	
	Average value of *a*	0.40709		
ST	*d*	0.23529	0.20362	5.33
	*a*	0.40753	0.40724	
	Average value of *a*	0.40739		
ST + AA	*d*	0.23495	0.20337	4.15
	*a*	0.40694	0.40675	
	Average value of *a*	0.40685		

## Data Availability

The data that support the findings of this study are available from the corresponding author (L.J.) upon reasonable request.

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
