# Peer review of "Effects of Heat Treatment on Microstructure and Mechanical Properties of Weldable Al–Mg–Zn–Sc Alloy with High Strength and Ductility"

_materials, 2023, doi:10.3390/ma16155435_

Round 1
Reviewer 1 Report
Dear authors,
1. Poor image quality
2. The inscriptions on the drawings are fuzzy, difficult to read
3. Figure 3 - high-resolution transmission electron microscopy - I couldn't find it in the images
4. Figure 7 - why do the engineering stress-strain curves look like saw? Did you stop the experiment and measure the geometric dimensions of the sample? I ask for clarification on the text of the article.
Reviewer 2 Report
1) The introduction of the manuscript should be focused on the novelty of the work. Add more sentences and recent references to justify novelty.
2) Fig. 1: Temperature range should be mentioned in the legend.
3) Fig. 2: Poor image quality and scale bar is not visible.
4) Fig. 3, 4 and 5: Poor image quality. Text inside the figure, table and scale bar are not visible.
5) Fig. 6: Peaks should be matched with the identified phase and phases should be written on the peak.
Minor editing of English language required
Reviewer 3 Report
The authors performed interesting experiments and calculations to elucidate the strengthening mechanisms of Al-Mg-Zn-Sc alloy for three treatment variants: induction casting, solid-solution treatment and artificial aging process. The research methods applied and the equipment used are adequate for the intended purposes.
However, the presentation of the results needs major improvement to convince the reader of the correctness of the conclusions.
- The illustrative material is not very clear, the scales on the microscopic images (both MO and SEM) are poorly visible. The magnifications adopted in Figure 3 do not allow a clear indication of the differences in the microstructure of the ST and ST+AA samples.
- The results of the EDS analysis in Figures 3, 4 and 5 are unreadable. The distinction between bean-like and square-like morphology (Figure 4, description - p. 6, line 157-158) is wishful thinking, at least in the images presented.
- Figure 6 is worth presenting in a narrow-angle version to actually show the differences in peak position. The whole tested range 2Q from 20 to 90 degrees adds little.
- The schematic diagram in Figure 8 should be more extensively explained: is the amount of nano Al3Sc precipitates similar in all three samples? Where is the zinc in the as-cast sample? Is the zinc in the ST sample in the form of a solid solution or nano-solutions, as the red dots would indicate? Did only the T-Mg32(Al, Zn)49 phase separate during ageing?
- On what basis was the volumetric contribution of the phases nano-precipitations to the microstructure assessed?
- The very small effect of heat treatment on the mechanical properties of the alloy is also questionable. A significant difference can only be seen in the increase in elongation after ST and AA processes compared to the cast sample, which can be explained by the dissolution of precipitates of intermetallic phases located in large numbers at grain boundaries in the as-cast sample. In contrast, the lack of strengthening of the material as a result of the ageing process is surprising. It is not stated on how many samples the strength tests were carried out.
- Perhaps it would be worth showing the correlation between calculated and experimental data?
- Editorial error:
-- in the description of Fig. 2 it is: (b,e) ST+AA sample, should be: (c,f);
--what does the Al3S(Sc,Ti) phase mean in the description in Figure 8?
